# Sampling the conformational space of the catalytic subunit of human γ-secretase

Xiao-chen Bai[1], Eeson Rajendra[1], Guanghui Yang[2], Yigong Shi[2]*, Sjors HW Scheres[1]*

[1]MRC Laboratory of Molecular Biology, Cambridge, United Kingdom; [2]Ministry of Education Key Laboratory of Protein Science, Tsinghua-Peking Joint Center for Life Sciences, Center for Structural Biology, School of Life Sciences, Tsinghua University, Beijing, China

**Abstract** Human γ-secretase is an intra-membrane protease that cleaves many different substrates. Aberrant cleavage of Notch is implicated in cancer, while abnormalities in cutting amyloid precursor protein lead to Alzheimer's disease. Our previous cryo-EM structure of γ-secretase revealed considerable disorder in its catalytic subunit presenilin. Here, we describe an image classification procedure that characterizes molecular plasticity at the secondary structure level, and apply this method to identify three distinct conformations in our previous sample. In one of these conformations, an additional transmembrane helix is visible that cannot be attributed to the known components of γ-secretase. In addition, we present a γ-secretase structure in complex with the dipeptidic inhibitor N-[N-(3,5-difluorophenacetyl)-L-alanyl]-S-phenylglycine t-butyl ester (DAPT). Our results reveal how conformational mobility in the second and sixth transmembrane helices of presenilin is greatly reduced upon binding of DAPT or the additional helix, and form the basis for a new model of how substrate enters the transmembrane domain.

*For correspondence: shi-lab@ tsinghua.edu.cn (YS); scheres@ mrc-lmb.cam.ac.uk (SHS)

## Introduction

γ-Secretase clears the anchors of type-I membrane proteins that are left behind in the membrane after shedding of their ectodomain. The substrate specificity of this intra-membrane protease is remarkably relaxed. It will cleave a wide range of substrates, as long as they form a single hydrophobic transmembrane helix, and the remaining ectodomain is not too large (*Struhl and Adachi, 2000*). As a manifestation of its promiscuous substrate specificity, γ-secretase also cleaves its substrate in different positions. For two of its most studied substrates, the Notch receptor and the amyloid precursor protein (APP), γ-secretase performs an initial endopeptidase-like ε-cleavage, which is followed by carboxypeptidase-like trimming, called γ-cleavage. Cleavage of APP leads to secretion of β-amyloid (Aβ) peptides into the extracellular environment. Abundant deposits of Aβ peptides in the brain, clinically known as β-amyloid plaques, are a defining characteristic of Alzheimer's disease (AD). Variability in the position of both ε– and γ-cleavages results in Aβ peptides with lengths ranging from 36 to 49 residues, with Aβ40 being the most common form. Longer peptides seem to be more prone to aggregation, and increased ratios of Aβ42/Aβ40 are thought to play a role in AD pathogenesis (*Tanzi and Bertram, 2005*).

Because of its central role in Aβ generation, γ-secretase is an attractive target for treatment of AD. However, a clinical trial with the γ-secretase inhibitor semagacestat had to be interrupted prematurely due to strong side effects, including skin cancer, weight loss and a faster decline of the cognitive skills of patients receiving the highest dose of the drug (*De Strooper, 2014*). Probably, global inhibition of the complex to reduce the formation of Aβ-peptides is undesirable, as this may also affect other pathways such as Notch signaling. Therefore, the development of modulators of γ-

**eLife digest** An enzyme called gamma-secretase cuts other proteins in cells into smaller pieces. Like most enzymes, gamma-secretase is expected to move through several different three-dimensional shapes to perform its role, and identifying these structures could help us to understand how the enzyme works.

One of the proteins that is targeted by gamma-secretase is called amyloid precursor protein, and cutting this protein results in the formation of so-called amyloid-beta peptides. When gamma-secretase doesn't work properly, these amyloid-beta peptides can accumulate in the brain and large accumulations of these peptides have been observed in the brains of patients with Alzheimer's disease. Earlier in 2015, a group of researchers used a technique called cryo-electron microscopy (cryo-EM) to produce a three-dimensional model of gamma-secretase. This revealed that the active site of the enzyme, that is, the region that is used to cut the other proteins, is particularly flexible.

Now, Bai et al. – including many of the researchers from the earlier work – studied this flexibility in more detail. For the experiments, gamma-secretase was exposed to an inhibitor molecule that stopped it from cutting other proteins. This meant that the structure of gamma-secretase became more rigid than normal, which made it possible to collect more detailed structural information using cryo-EM. Bai et al. also developed new methods for processing images to separate the images of individual enzyme molecules based on the different shapes they had adopted at the time. These methods make it possible to view a mixture of very similar enzyme structures that differ only in a small region of the protein (in this case the active site).

In the future, it would be useful to repeat these imaging experiments using a range of different molecules that alter the activity of gamma-secretase. Furthermore, the new image processing methods developed by Bai et al. could be used to study flexibility in the shapes of other proteins.

secretase activity that leave ε-cleavage intact but stimulate γ-cleavage has been suggested as an alternative (*Wolfe, 2012*). Moreover, the development of specific inhibitors of Notch cleavage may be beneficial for the treatment of cancer. However, such developments are currently hindered by a lack of quantitative insights into the mechanism of γ-secretase proteolysis.

The γ-secretase complex consists of four essential, integral membrane proteins: presenilin (PS), nicastrin, anterior pharynx defective 1 (Aph-1), and presenilin enhancer 2 (Pen-2) (*De Strooper, 2003*; *Kimberly et al., 2003*). PS provides the two essential aspartates that form the proteolytic active site where both ε and γ-cleavages occur. More than 200 missense mutations in the gene for PS have been linked to an early-onset, familial form of Alzheimer's disease (FAD) ((*De Strooper et al., 2012*) and http://www.alzforum.org/mutations). In humans, two genes encode two homologous forms of PS (PS1 and PS2), but most FAD-derived mutations target PS1. Because there are also two forms of Aph-1 (Aph-1a and Aph-1b), four possible complexes may form, of which nicastrin: Aph-1a:PS1:Pen-2 is the most abundant. Nicastrin has a large, heavily glycosylated ectodomain, which is thought to play a role in substrate recognition (*Shah et al., 2005*). Aph-1 has been proposed to play a scaffolding role (*Lee et al., 2004*). Pen-2 is essential for proteolytic activity of the mature complex, and facilitates auto-proteolysis of PS1 in a long cytosolic loop between its sixth and seventh transmembrane helices (TM6 and TM7) (*Thinakaran et al., 1996*).

We recently solved the cryo-EM structure of a nicastrin:Aph-1a:PS1:Pen-2 complex to a resolution of 3.4 Å (*Bai et al., 2015a*). This structure allowed building a near-complete atomic model, and revealed how Aph-1 and Pen-2 hold a remarkably flexible PS1 subunit underneath the nicastrin ectodomain. In particular, TM2, the cytoplasmic side of TM6 and the long linker between TM6 and TM7 of PS1 were largely disordered. The presence of the linker could only be inferred from fuzzy densities in 2D class averages, whereas the approximate position of TM2 could only be inferred from a 7 Å low-pass filtered map. This low-pass filtered map also showed a rod-shaped density in the cavity between TM2, TM3 and TM5, but the identity of this density could not be determined. Although the sample used for structure determination did show proteolytic activity for APP (*Lu et al., 2014*), the active site appeared to be in an inactive conformation as the two catalytic aspartates were too far apart to catalyze proteolysis.

In this paper, we set out to gain further insights into the proteolytic mechanism of γ-secretase by using two complementary approaches to sample the conformational landscape of its catalytic subunit. With the aim of trapping the complex in a more defined conformation, we solve a structure in complex with the non-transition-state analogue inhibitor DAPT, which is a precursor in the development of semagacestat (*Dovey et al., 2001*). In addition, we describe how masked cryo-EM image classification combined with subtraction of part of the signal from the experimental images allows visualizing molecular dynamics of the catalytic subunit in its apo-state at the secondary structure level. The resulting four structures represent a significant step towards understanding how this protease cleaves its many substrates.

## Results

### Approach

A powerful method of dealing with structural heterogeneity in cryo-EM data sets is to 'focus' refinement on a defined region of the protein complex of interest. In this approach one masks out part of the reference during 3D refinement, thereby effectively ignoring structural variability in less interesting parts. For example, we used masked refinements to deal with variability in the relative orientations of ribosomal subunits (*Amunts et al., 2014*; *Wong et al., 2014*). Similarly, masked multi-reference refinement may be used as a clustering tool, i.e. to separate experimental particle images based on differences in a defined region of interest. We refer to this approach as masked 3D classification. However, masked 3D classifications aimed at analyzing the conformational landscape of γ-secretase in its apo-state were unsuccessful. An initial data set of 400,000 particles gave rise to a 3.5 Å map. Using different masks on the transmembrane domain, masked 3D classification consistently yielded only a single class showing good density. Although this approach did result in the selection of 160,000 particles from which we could calculate a better 3.4 Å map, it did not reveal the nature of conformational freedom within the catalytic subunit (*Bai et al., 2015a*).

A fundamental problem with masked refinements is that one compares projections of a partial map with experimental projections of the entire particle (*Figure 1*). This leads to inconsistencies in the comparisons that underlie the refinement procedure. For example, one might want to focus classification on the part of the particle that is depicted in red in *Figure 1A*, and ignore any variations in the yellow part. Masking away the yellow part from the reconstruction in masked refinements (*Figure 1C*) yields reference projections that only contain the red part (*Figure 1F*). However, each experimental image (*Figure 1D*) contains signal coming from the entire particle, *i.e.* from both the yellow and the red parts. Therefore, the yellow part of the signal will act as an additional source of noise in the comparison between the experimental image and the masked reference projection. It will depend on the signal-to-noise ratio in the original image and on the size of the part of the complex that is masked away, whether this additional noise will affect the refinement. For large particles, high signal-to-noise ratios in the data make masked refinements relatively robust, but even for ribosomes masked refinements of the small subunit proved much more difficult than for the large subunit (*Wong et al., 2014*).

To reduce the inconsistency in image comparison, we explored a modification of the masked classification approach. Using the example in *Figure 1*, if the noisy experimental image were to contain only signal from the red part, then a masked refinement would be consistent. To emulate this situation, we subtract projections of the yellow part of the reconstruction (*Figure 1B,E*) from every experimental image. This requires knowledge about the relative orientation of each particle, which is obtained from a consensus refinement of the entire data set against a single, unmasked reference. Also, we apply the CTF of each experimental image to the projection of the yellow part prior to the subtraction. The resulting, modified experimental particle image (*Figure 1G*) then ideally only contains the signal from the red part of the particle, and the only inconsistency in the image comparison is the original experimental noise. Therefore, using the subtracted experimental particle images in masked refinements that are focused on the red part of the signal will be better than using the original images.

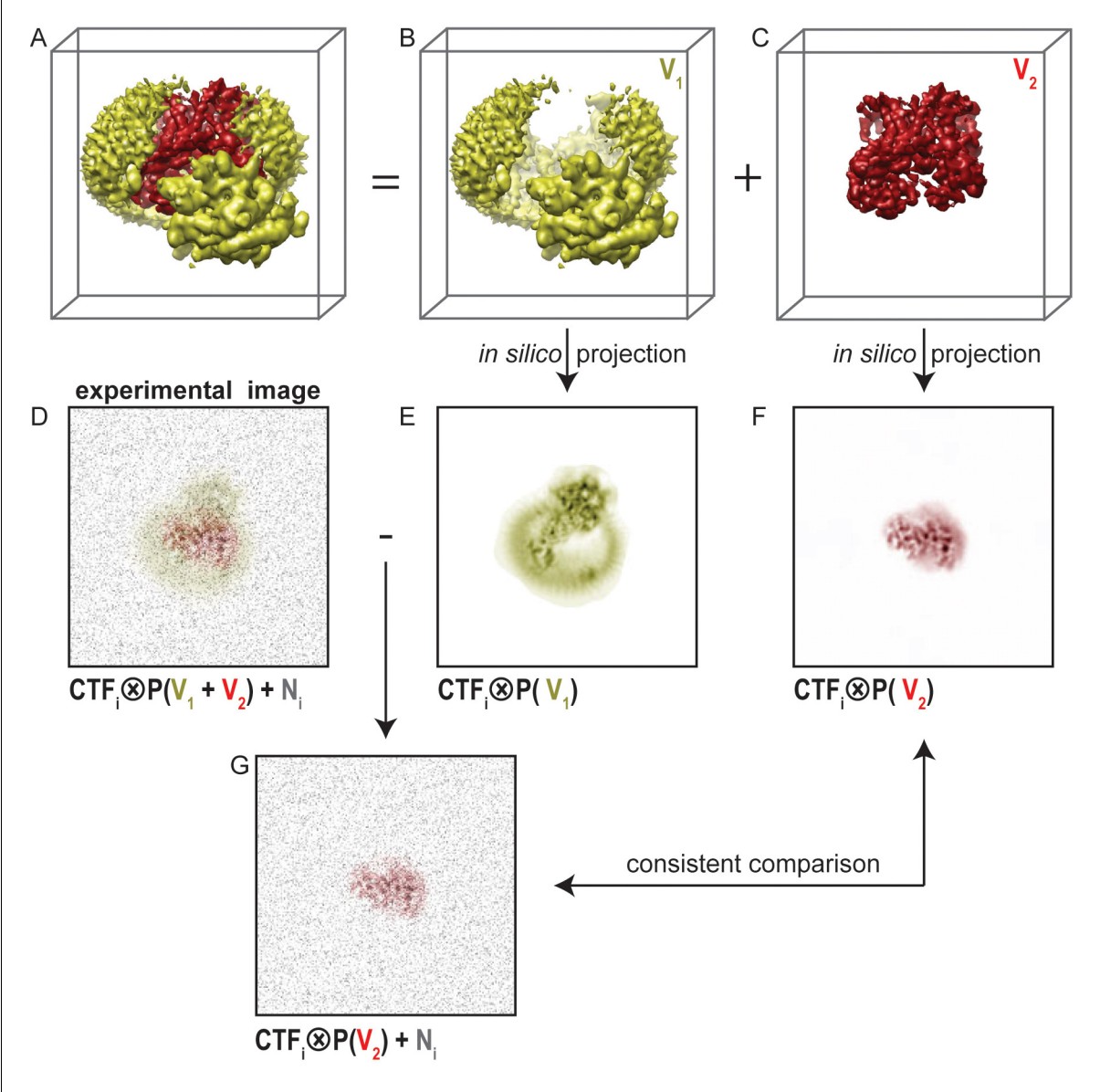

**Figure 1.** Masked classification with residual signal subtraction. (**A**) A 3D model of a complex of interest. (**B**) The part of the complex one would like to ignore in masked classification ($V_1$) is shown in yellow. (**C**) The part of the complex one would like to focus classification on ($V_2$) is shown in red. (**D**) An experimental particle image is assumed to be a 2D projection of the entire complex in panel A that is affected by the contrast transfer function (CTF), and to which experimental noise (N, shown in grey) has been added. (**E**) A CTF-affected 2D projection of $V_1$. (**F**) A CTF-affected 2D projection of $V_2$. Previous approaches to masked classification in RELION (*Amunts et al., 2014*; *Wong et al., 2014*) compared experimental particles (panel D), with reference projections of only $V_2$ (panel F). This results in inconsistent comparisons. (**G**) In the modified masked classification approach, one subtracts the CTF-affected 2D projection of $V_1$ (panel E) from the experimental particle (panel D). This results in a modified experimental particle image that only contains experimental noise and the CTF-affected projection of $V_2$, so that comparison with the reference projection in panel F becomes consistent.

## Masked classification with signal subtraction on the apo-state data set

Because the PS1 subunit showed the highest level of disorder in our high-resolution structure, we decided to perform masked classification on PS1 with subtraction of the signal from the rest of the complex. Since the total molecular weight of the ordered part of PS1 in our previous map was less than 30 kDa, we reasoned that the remaining signal in the subtracted experimental images would probably not be strong enough to allow the determination of their relative orientations. Therefore, we performed masked classification on the set of 400,000 particles, while keeping all orientations

fixed at the values determined in the refinement of the 3.5 Å consensus map. Classification into eight classes yielded three majority classes that showed good density (*Figure 2*). Five smaller classes gave suboptimal reconstructions, and the particles from these classes were discarded. The (original, non-subtracted) particles from the good classes were then subjected to separate 3D auto-refinement runs (*Scheres, 2012*), all of which were started from the same 40 Å low-pass filtered reference to avoid model bias. The resulting maps were very similar in the nicastrin and Aph-1 subunits, but obvious differences were present in the PS1 and Pen-2 subunits. The maps had resolutions in the range of 4.0–4.3 Å, which allowed reliable main-chain tracing, but left density for many side chains less well defined (*Table 1*, *Figure 3*, *Figure 3—figure supplement 1*).

As a control for model bias we performed a cross-refinement, where the 10 Å low-pass filtered map from class 3 was used as initial reference for refinement of the particles assigned to class 1 (*Figure 2—figure supplement 1*). The initial reference did not influence convergence, as the cross-refinement yielded a map that was indistinguishable from the one obtained for class 1 in *Figure 3*. As a negative control, we performed masked classification with signal subtraction using eight classes on the Aph-1 subunit (*Figure 2—figure supplement 2*). The density for this subunit in the consensus map was very well defined, indicating that this subunit is much more rigid than PS1. In this case, the eight resulting classes attracted similar numbers of particles and all eight classes gave rise to very similar reconstructions. Finally, in a third control to test reproducibility we performed multiple different masked classifications on the PS1 subunit (*Figure 2—figure supplement 3*). We used different numbers of classes (six and ten instead of eight), a different random seed, or slightly different masks on PS1. In all cases, although the class distributions and the structural details varied, the resulting classes revealed similar differences for TM2 and TM6. In addition, we tested our method on a simulated set of images containing a mixture of projections from the maps of classes 1 and 2 in *Figure 3*. For these simulations we used similar signal-to-noise ratios, CTF parameters and orientational distributions as observed in our experimental data set (also see Methods). Masked classification with signal subtraction on the PS1 subunit correctly identified 93% of the simulated particles (*Figure 2—figure supplement 4*).

Comparison of the three structures that were identified in the experimental data set using masked classification on the PS1 subunit (*Table 1*, *Videos 1–2*) explained observations made in the consensus structure. In the high-resolution structure, density for the cytoplasmic side of TM6 was weak, while density for TM2 was only visible after applying a 7 Å low-pass filter. This agrees with the observation that TM2 is only ordered in class 1, whereas TM6 adopts different orientations in all three classes. Control classifications with variations in the number of classes, random seeds or masks revealed even more variations in the conformation of TM6 (e.g. see *Figure 2—figure supplement 3*). Probably, TM6 adopts a wide range of conformations in solution and our classification merely provides discrete snapshots of a continuum.

Despite the fact that we did not focus our classification on Pen-2, we also observe significant variations in the conformation of Pen-2. Compared to classes 1 and 2, Pen-2 in class 3 has rotated away from PS1. This rotation concurs with a large conformational change in PS1, where TM3 and TM4 rotate in the same direction as Pen-2, TM5 and TM6 move towards the extracellular space, and TM6 rotates towards TM7 (see *Video 2*). The rotation and upward motion of TM6 positions the two catalytic aspartates within close enough distance of each other to potentially catalyze proteolysis (although we do not see density for the aspartate side chains).

When the high-resolution consensus map was low-pass filtered to 7 Å, besides density for TM2 a second, unidentified rod-shaped density was also observed in the cavity formed by TM2, TM3 and TM5 of PS1 (*Bai et al., 2015a*). A similar density that could not be attributed to any of the known γ-secretase components is also visible in classes 1 and 2, but not in class 3. The rod-shaped density is best defined in class 1, where it shows clear features of α-helical pitch. We modeled this density as an α-helix with an almost 90-degree kink at the extracellular side of the transmembrane domain (*Figure 4*). The kink of this helix is in close proximity of the loop of residues 240–244 of nicastrin, and then extends into the membrane through the cavity formed by TM2, TM3 and TM5 of PS1, until it disappears just before reaching the active site. The density closest to the active site looks less helical, and in this region we modeled it as an extended chain. The entire cavity in which the helix is present is lined with residues from TM2, TM3 and TM5 of PS1 that have been implicated in FAD. The cryo-EM density was not of sufficient quality to allow the identification of this helix. Mass spectrometry analysis suggested the presence of multiple proteins with a single predicted

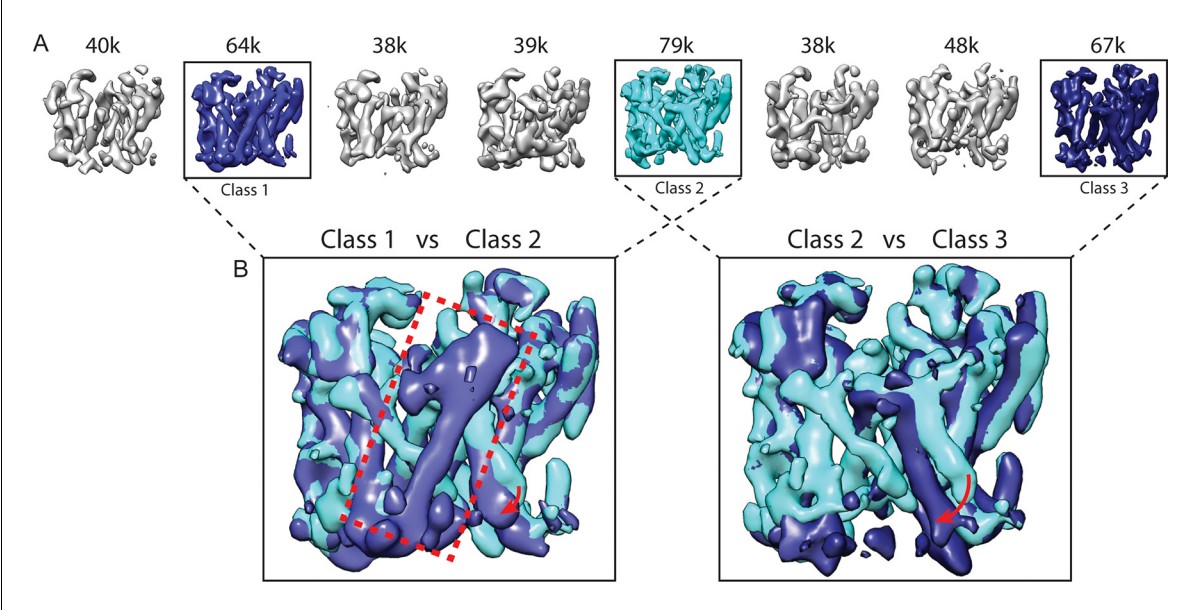

**Figure 2.** Masked classification with signal subtraction on PS1. (**A**) From a masked 3D classification run on the PS1 subunit, the three largest classes (in cyan and blue, labeled class 1–3) showed good density. The five smallest classes (in grey) were ignored in further analyses as they showed suboptimal density. (**B**) Superposition of classes 1 and 2 and of classes 2 and 3 reveals the different orientations of TM6 in all three classes (indicated with red arrows), and the fact that TM2 (indicated with a red dashed box) is only ordered in class 1.

The following figure supplements are available for figure 2:

**Figure supplement 1.** Cross-refinement of the masked classification results.

**Figure supplement 2.** Masked classification on Aph-1.

**Figure supplement 3.** Reproducibility of masked classification on PS1.

**Figure supplement 4.** Masked classification with simulated data.

transmembrane helix in our sample, and the presence of three of these proteins was further confirmed by Western blot analysis (*Figure 4—figure supplement 1*).

## Binding of DAPT rigidifies the catalytic subunit

In order to gain further insights into the plasticity of the catalytic subunit, we also performed cryo-EM single-particle analysis on γ-secretase in complex with DAPT (*Figure 5*, *Figure 5—figure supplement 1*). Despite collecting a comparable amount of micrographs as we did for the apo-state complex, 2D and 3D classification approaches selected less than 20% of the complexes as suitable for high-resolution reconstruction. This contrasts with a selection of approximately 40% for the apo-state data set. Because the overall appearance of the micrographs for both data sets was similar, it could be that either DAPT or the dimethyl sulfoxide (DMSO) in which DAPT was dissolved interfered with the structural integrity of the complex. Nonetheless, from the selected 51,366 particles we calculated a 4.2 Å map, which was of sufficient quality to build a reliable main-chain model, although the density for many side chains was less clear. For these data, masked classifications with signal subtraction revealed only a single class with good density for the transmembrane helices, and this class did not show any helical-like density in the cavity between TM2, TM3 and TM5 of PS1 (*Figure 5—figure supplement 2*).

Upon inhibitor binding, there are no prominent changes in Aph-1 and nicastrin, and Pen-2 is in a very similar conformation as in classes 1 and 2 of the apo-state ensemble. The largest changes occur in PS1, which is much better ordered in the complex with DAPT. TM2 and its linkers with TM1 and

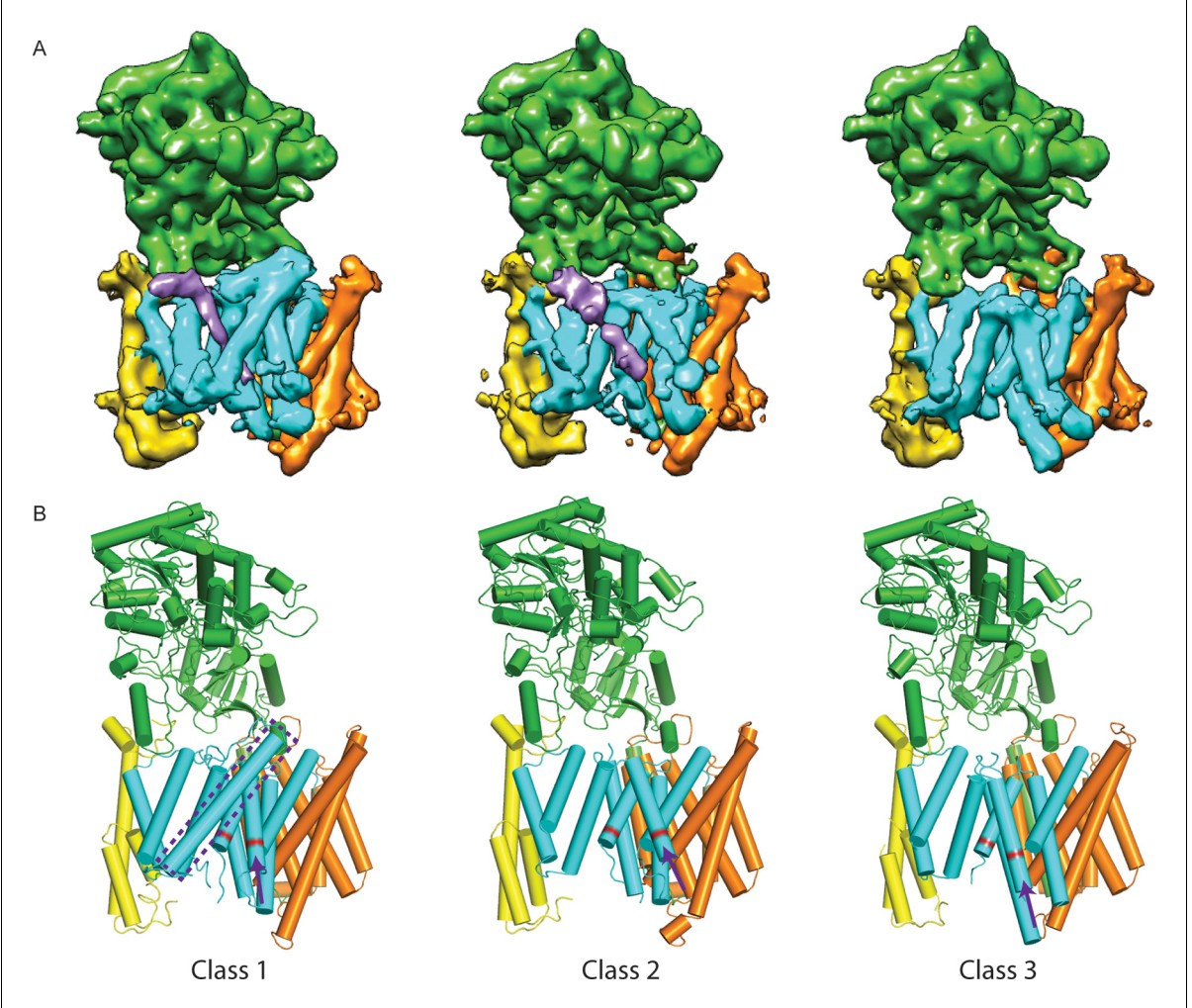

**Figure 3.** Three classes from the apo-state ensemble. (**A**) The reconstructed density for the three classes. Nicastrin is shown in green, Aph-1 in orange, PS1 in cyan, and Pen-2 in yellow. α-Helical density that is unaccounted for by the γ-secretase model is shown in purple. The same color code is used throughout this paper. (**B**) Schematic representation of the γ-secretase atomic models. TMs of PS1 are numbered. The active site aspartates are shown in red. The purple dashed box highlights TM2 of PS1, which is only ordered in class 1. The purple arrows indicate the orientation of TM6, which is different in each class.

The following figure supplement is available for figure 3:

**Figure supplement 1.** Fourier shell correlations for the three apo-state classes.

TM3, the cytoplasmic ends of TM3 and TM6, and part of the long linker between TM6-7 all become ordered upon DAPT binding (*Figure 5—figure supplement 3*). The linker between TM1 and TM2 sticks partially into the transmembrane region to fold back out again to connect to TM2. This helix contains 32 residues and runs at an angle of approximately 40 degrees with the plane of the membrane. At the cytoplasmic side of the membrane a short linker connects TM2 to TM3, and TM3 extends 7 residues longer than in the apo-state. Interestingly, apart from local changes in the cytoplasmic sides of TM2 and TM6, the overall conformation of PS1 in the DAPT-bound structure is very similar to class 1 from the apo-state ensemble, and the two structures overlap with a root mean squared deviation (r.m.s.d.) of 0.4 Å between 265 Cα atom pairs (*Figure 5—figure supplement 4*).

Near the active site, TM6 displays a strong kink at Pro264, which positions its cytoplasmic side underneath a cavity formed by TM2, TM3, TM5, TM6 and TM7. This cavity contains the only peak of strong density in the transmembrane domain that is unaccounted for by the protein model

**Table 1.** Refinement and model statistics

| | Class1 | Class2 | Class3 | DAPT |
|---|---|---|---|---|
| **Data collection** | | | | |
| Particles | 63,873 | 79,263 | 66,720 | 51,366 |
| Pixel size (Å) | 1.4 | 1.4 | 1.4 | 1.4 |
| Defocus range (μm) | 0.7–3.2 | 0.7–3.2 | 0.7–3.2 | 0.6–2.8 |
| Voltage (kV) | 300 | 300 | 300 | 300 |
| Electron dose (e-/Å$^{-2}$) | 40 | 40 | 40 | 40 |
| **Map features** | | | | |
| Density TM2 | + | - | - | + |
| Cα-Cα distance D257– D385 (Å) | 9.5 | 12.7 | 9.1 | 8.0 |
| Conformation Pen-2 | in | in | out | in |
| α-helical density | + | + | - | - |
| **Model composition** | | | | |
| Non-hydrogen atoms | 10,443 | 9,922 | 9,916 | 10,543 |
| Protein residues | 1,315 | 1,245 | 1,247 | 1,329 |
| **Refinement** | | | | |
| Resolution (Å) | 4.1 | 4.0 | 4.3 | 4.2 |
| Map sharpening B-factor (Å$^2$) | −100 | −100 | −130 | −130 |
| Fourier Shell Correlation | 0.8236 | 0.8818 | 0.8050 | 0.8602 |
| Rfactor | 0.2917 | 0.3028 | 0.2816 | 0.3426 |
| **Rms deviations** | | | | |
| Bonds (Å) | 0.0112 | 0.0095 | 0.0120 | 0.0093 |
| Angles (°) | 1.7352 | 1.6083 | 1.7922 | 1.6186 |
| **Model geometry** | | | | |
| Molprobity score | 3.17 | 2.89 | 3.16 | 3.22 |
| Clashscore (all atoms) | 17.35 | 9.31 | 13.26 | 18.15 |
| Good rotamers (%) | 89.6 | 90.9 | 86.6 | 88.9 |
| **Ramachandran plot** | | | | |
| Favored (%) | 85.1 | 84.4 | 84.2 | 84.3 |
| Allowed (%) | 10.9 | 12.3 | 11.5 | 11.8 |
| Outliers (%) | 4.0 | 3.3 | 4.3 | 3.9 |

(*Figure 5C*). It breaks up into disconnected pieces at 4.5 standard deviations above the mean, which is somewhat weaker than the surrounding transmembrane helices in PS1, which break up at approximately 6 standard deviations above the mean. The size of this density is consistent with the expected size of a single DAPT molecule. Therefore, we tentatively assign this density to the inhibitor, although at the limited resolution of our map we cannot determine its exact orientation or conformation. The pocket where the inhibitor binds is very hydrophobic, which is as anticipated given the hydrophobic nature of the DAPT molecule. In particular, Met146 and Met233, as well as Trp165, Phe283 and Gly384 seem to be involved in interactions with the inhibitor. Except for Phe283, all these residues have been targeted for mutations in FAD patients. The location of the inhibitor right next to the active site is also in good agreement with previous observations that the DAPT binding site is distinct from, but in close proximity of the active site (*Kornilova et al., 2003*; *Morohashi et al., 2006*).

Combined with a small movement of TM7 towards TM6, the kink in TM6 brings the Cα atoms of the two catalytic aspartates within 8.0 Å of each other, which may again be close enough to facilitate

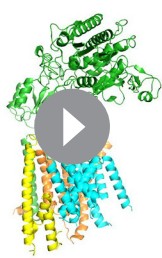

**Video 1.** A morph between the atomic models from class 1 and 2 of the apo-state ensemble.

catalysis. Also the conformation of the highly conserved [433]PAL[435] motif, which is important for γ-secretase activity (*Wang et al., 2004*, *2006*), changes upon DAPT binding. A large part of the linker between TM6 and TM7 remains invisible, but at the cytoplasmic end of TM6 part of this linker becomes ordered as it passes beneath the inhibitor (*Figure 5C*). There, residues 280-283 form a single α-helical turn, which is stabilized by a hydrogen bond between Glu280 and His163, while Phe283 interacts with the inhibitor. Mutation of Glu280 into an alanine (the so-called Paisa mutation) is by far the most common cause of FAD (http://www.alzforum.org/mutations). Residues 285–288 form a short β-strand before the density of the linker disappears just three residues before the auto-proteolytic cleavage site between Thr291 and Met292. The density then reappears at residue 378, where residues 378–381 form a second β-strand that hydrogen bonds with the first.

## Discussion

Although many proteins employ functionally important flexibility at the level of secondary structure, few experimental techniques exist for the study of this dynamics. Nuclear magnetic resonance (NMR) is a powerful tool for the characterization of structural ensembles in dynamic complexes, but its applicability is typically limited to proteins with a molecular weight below 40– 50 kDa. For complexes larger than several hundred thousand daltons, dynamic changes in tertiary and quaternary structure have been studied by cryo-EM image classification, in particular with the recent advent of direct electron detectors and improved computer algorithms (*Bai et al., 2015b*; *Dashti et al., 2014*). However, for complexes that are too large for NMR, the characterization of molecular dynamics within individual protein domains has typically been restricted to computer simulations. The procedure for masked cryo-EM image classification combined with residual signal subtraction fills part of this gap in experimental techniques. The idea to subtract part of the signal from experimental cryo-EM images is not new. Michael Radermacher and colleagues subtracted partial projections from cryo-EM images to study symmetry mismatches in bacteriophage φ29 (*Morais et al., 2003*) and flaviviruses (*Zhang et al., 2007*); Steven Ludtke and colleagues used image subtraction in the *e2ligandclassify.py* program to separate ribosomes with and without secY channels (*Park et al., 2014*); Hongwei Wang and colleagues used a modified version of RELION to subtract projections of NSF rings to analyze a symmetry mismatch and structural variability in the SNAP-SNARE complex (*Zhou et al., 2015*); we used an iterative image subtraction method in RELION to improve the density of a flexible domain of the spliceosomal U4/U6.U5 tri-snRNP complex (*Nguyen et al., 2015*); and most recently, Huiskonen and colleagues also used RELION to subtract viral capsid densities in order to visualize an RNA polymerase bound inside the virus (*Ilca et al., 2015*). The procedure described here provides an easily accessible and generally applicable tool for signal subtraction coupled to masked refinements and/or classifications of single-particle data. Its successful application to γ-secretase demonstrates its potential for complexes that are considered to be relatively small for cryo-EM structure determination, and shows that it allows separation of protein structures that differ only in the orientation and position of a few α-helices.

Application of the masked classification approach to the data set of apo-state γ-secretase particles revealed a range of different conformations for TM6. This conformational

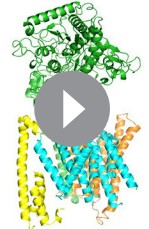

**Video 2.** A morph between the atomic models from class 2 and 3 of the apo-state ensemble.

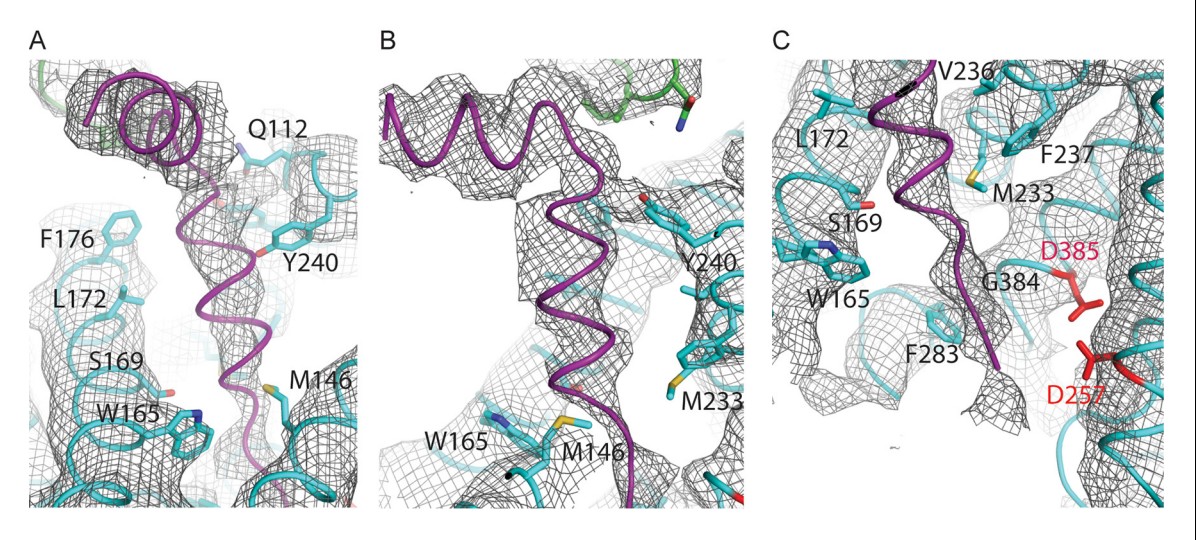

**Figure 4.** Helical density in class 1 of the apo-state ensemble. (**A–C**) Three different views of the map and the atomic model are shown. The kinked α-helix that is unaccounted for by the γ-secretase model is shown in purple. PS1 residues that interact with this helix are labeled.

The following figure supplement is available for figure 4:

**Figure supplement 1.** Mass spectrometry and Western blot analyses.

flexibility leads to a variation in the distance between the two aspartates that form the active site. Interestingly, the class where the aspartates are closest together also shows a markedly different conformation of Pen-2 and a re-arrangement of TM3, TM4 and TM5 in PS1. Pen-2 is required for autocatalytic maturation and protease activity of γ-secretase. Binding of Pen-2 to the complex activates the active site, and binding of Pen-2 was observed to have an allosteric effect on TM6 (*Takeo et al., 2012*). However, in the absence of PS1 structures without Pen-2 bound, it will probably remain unclear whether the changes in Pen-2 and PS1 observed here are relevant to this allosteric activation mechanism.

Our analysis of the structure in complex with DAPT provides complementary insights into the conformational freedom of γ-secretase. Upon binding of the inhibitor, the catalytic subunit undergoes a marked rigidification. TM2 and its linkers to TM1 and TM3 become ordered, and so do the cytoplasmic half of TM6 and part of the linker between TM6 and TM7. Around the active site, the conformations of the kink in TM6 and the conformation of the long linker between TM6 and TM7 are noticeably different from the apo-state consensus structure. Maturation of the γ-secretase complex requires auto-proteolytic cleavage at Thr290 (or alternatively at Val292 or Met297). The observed kink in TM6 may expedite the U-turn that is required to position the auto-proteolytic cleavage site back into the active site. Moreover, auto-proteolytic cleavage is predicted to occur in an α-helix spanning residues 280–300 of the linker. In the DAPT-bound structure, a single helical turn starts at Glu280, but residues 285–288 form a short β-sheet with the end of the linker that connects to TM7. The auto-proteolytic site is still flexible in this structure, as the density disappears after Tyr288. Since auto-proteolysis appeared to be complete in our sample (*Lu et al., 2014*), it could be that residues 280–300 do form a helical structure prior to self-cleavage. Alternatively, it could be that the inhibitor specifically alters the secondary structure in the linker, for example through its observed interaction with Phe283.

TM2 of PS1 is well ordered in both the DAPT-bound structure and in class 1 of the apo-state, whereas it is invisible in the other apo-state classes. This helix only seems to be ordered when something is bound in the large cavity that is lined with FAD-derived mutations between TM2, TM3 and TM5. In the inhibitor-bound structure this cavity contains the density that we attribute to DAPT, while density for a kinked α-helix is visible in class 1 of the apo-state. Although the density for the kinked α-helix was not of sufficient quality for unambiguous identification, mass spectrometry and

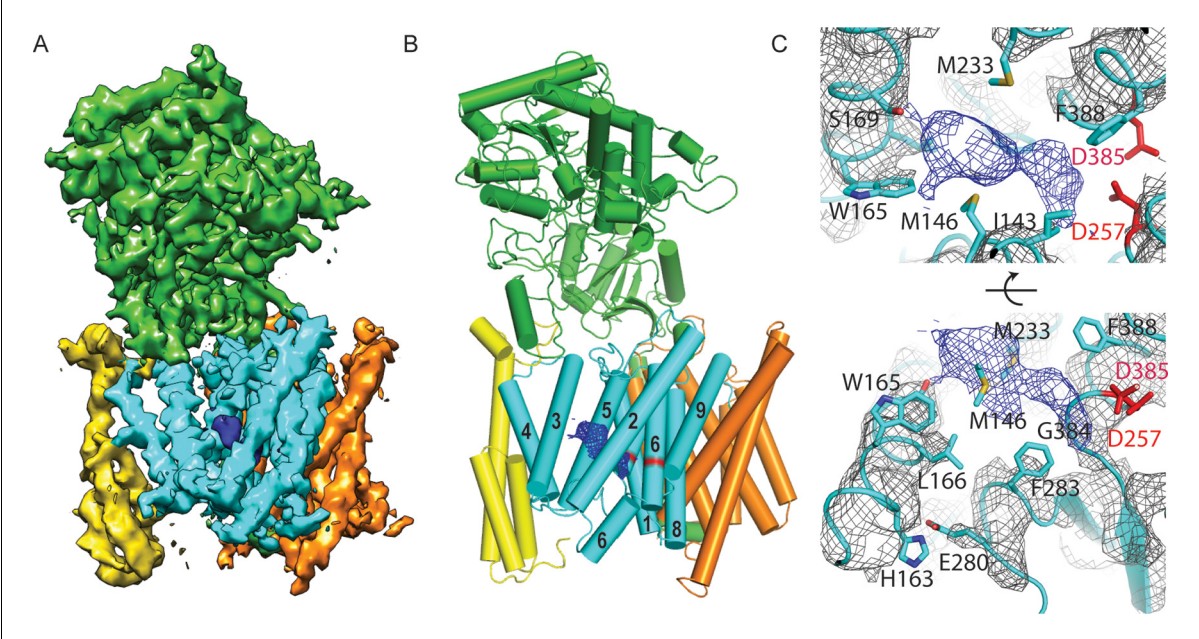

**Figure 5.** Cryo-EM structure of γ-secretase in complex with DAPT. (**A**) The reconstructed density for the entire complex. Density attributed to DAPT is shown in blue. (**B**) Schematic representation of the atomic model. TMs of PS1 are numbered. (**C**) Two approximately orthogonal close-ups of the DAPT-binding site. Residues that interact with DAPT, as well as His163 and Glu280 are labeled.

The following figure supplements are available for figure 5:

**Figure supplement 1.** Fourier shell correlations for the DAPT-bound structure.

**Figure supplement 2.** Masked classification with signal subtraction on the DAPT-bound data.

**Figure supplement 3.** Newly ordered elements in the DAPT-bound structure.

**Figure supplement 4.** Similarity between the DAPT-bound structure of PS1 and class 1.

Western blot analyses suggest that a mixture of different proteins with a single transmembrane helix may be present in our sample. Four of the proteins that were identified by mass spectrometry had also previously been observed to co-purify with γ-secretase: TMP21/p23, p24a, Vamp-8 and Sec22B (*Wakabayashi et al., 2009*). TMP21 was also observed to be a component of the γ-secretase complex that acts as a negative regulator of γ-cleavage, while leaving ε-cleavage intact (*Chen et al., 2006*).

We hypothesize that the kinked α-helix in our structure arises from a mixture of co-purified proteins in our sample that bind to the γ-secretase complex in a manner that mimics substrate binding. This hypothesis is in good agreement with previous biochemical observations about an 'initial substrate-binding site' that is distinct from the active site (*Beher et al., 2003*; *Das et al., 2003*; *Esler et al., 2002*; *Tian et al., 2002*). Photolabeling experiments suggest that the initial substrate-binding site partially overlaps with the DAPT-binding site (*Kornilova et al., 2003*; *Morohashi et al., 2006*), and mutational analysis identified TM2 and TM6 to be involved in substrate binding (*Watanabe et al., 2010*). Experiments with photoaffinity probes based on α-helical substrate-like inhibitors showed that DAPT could not displace a 10-residue long helical probe, suggesting spatially separated binding sites of the substrate and the inhibitor. This however was not the case for a 13-residue long peptide, for which addition of DAPT led to a strong reduction in photolabeling. Because this peptide also prevented labeling of a transition-state mimicking photoprobe, the longer α-helical probe probably also interacts with the active site (*Kornilova et al., 2005*). Our hypothesis explains these data well. A superposition of the kinked α-helix on top of the DAPT structure shows

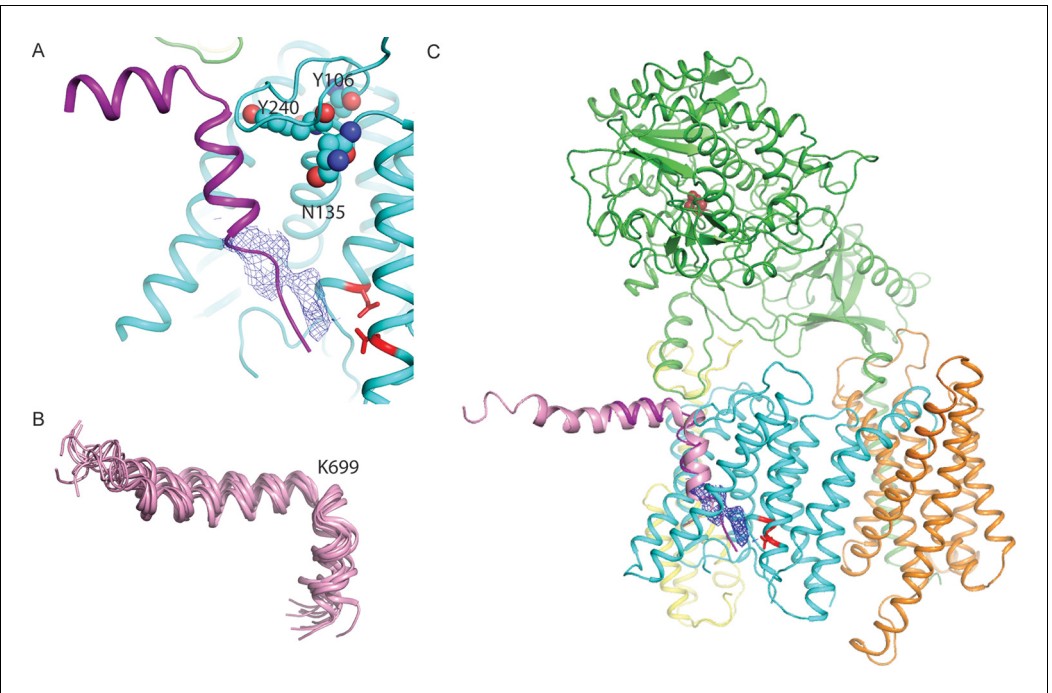

**Figure 6.** A hypothesis for substrate binding. (**A**) A superposition of the kinked α-helix from class 1 and the DAPT-density and atomic model for PS1 from the DAPT-bound structure. The lower end of the kinked helix and the DAPT density overlap. The recently identified residues that interact with a phenylimidazole-like γ-secretase modulator are shown with spheres. (**B**) Ensemble of NMR-models for an Aβ42 peptide in an aqueous solution of fluorinated alcohols. (**C**) Hypothetical model for how APP (one of the NMR models is shown in pink) binds to γ-secretase.

The following figure supplement is available for figure 6:

**Figure supplement 1.** Predicted transmembrane regions.

that after the kink, the α-helix extends for 10 residues into the transmembrane domain, before its four C-terminal residues overlap with the DAPT-binding site and almost reach the active site (*Figure 6A*).

Furthermore, the NMR structure of the Aβ42 peptide in an aqueous solution of fluorinated alcohols shows a strikingly similar 90-degree kink in its α-helical structure, which is positioned right after Lys699 (*Figure 6B*). A superposition of the NMR model on the kinked α-helix in our structure places its N-terminus flat on the extracellular side of the membrane, while the cleavage site that would result in the formation of Aβ42 peptides is placed in close proximity to the active site (*Figure 6C*). Interestingly, all the proteins that were both identified in our sample by mass spectrometry and that had also previously been observed to co-purify with γ-secretase (*Wakabayashi et al., 2009*) contain either an arginine or a lysine just before their predicted transmembrane helix (*Figure 6—figure supplement 1*). It is therefore tempting to speculate that the large, positively charged residue together with the kink in the α-helix act as an anchor to delimit the cleavage position in the substrate. This would explain why mutations in Lys699 have a marked effect on the length of the Aβ cleavage products (*Kukar et al., 2011*). In addition, the kink in the α-helix is positioned next to the recently mapped binding site of a phenylimidazole-like γ-secretase modulator (highlighted with spheres in *Figure 6A*), which could explain how this modulator affects cleavage in the distant active site through changes in the substrate conformation (*Takeo et al., 2014*). It could also explain why replacement of Lys699 or increasing the positive charge at Gly700 has been observed to block or attenuate the effects of γ-secretase modulators (*Jung et al., 2014*).

In conclusion, we show that masked cryo-EM image classification combined with the subtraction of part of the signal from the experimental images allows separation of structures that differ at the

secondary structure level. Application of this approach to our previously described data set of γ-secretase in its apo-state reveals distinct conformations for TM2 and TM6 of PS1. In one of the identified structures, we observe a kinked α-helix in the cavity between TM2, TM3 and TM5 of PS1 that cannot be attributed to any of the γ-secretase components. Mass spectrometry and Western blot analyses suggest the presence of a mixture of proteins with a single transmembrane helix that co-purified with the complex, one of which is a known modulator of γ-secretase cleavage. These results are complemented with a cryo-EM structure of the complex bound to the dipeptidic inhibitor DAPT. Binding of the inhibitor leads to a marked reduction in the flexibility of the catalytic subunit. Together, our results form the basis for a hypothesis that substrate enters the transmembrane domain through the cavity formed by TM2, TM3 and TM5 of PS1. The disordered nature of TM2 in the absence of substrate or inhibitor suggests that this helix may act as a lateral gate through which the transmembrane helix of the substrate enters the cavity. The observation that the inhibitor-bound structure closely resembles the class with the kinked α-helix suggests that the inhibitor and the substrate stabilize a similar conformation from the apo-state ensemble. DAPT would then act as an inhibitor by blocking access of the substrate to the active site. An appealing route to confirm our hypothesis, and to gain further mechanistic insights, is the determination of additional cryo-EM structures in complex with different substrates, substrate analogues, inhibitors or other modulators of activity. Masked classification with signal subtraction will be a useful tool in this endeavor, as understanding how different factors shift the equilibrium of conformational states in this flexible enzyme will be key to increase our understanding of its functioning.

## Materials and methods

### Electron microscopy

The sample preparation procedure and imaging conditions for the apo-state data were described previously (*Bai et al., 2015a*). To prepare the complex with DAPT, we incubated 20 μl of the same γ-secretase sample for 20 min at 4°C with 0.2 μl of a 10 mM solution of DAPT in 100% DMSO (yielding an approximate concentration of 4 μM γ-secretase, 100 μM DAPT and 1% DMSO). Subsequently, aliquots of 3 μl were applied to previously glow-discharged holey carbon grids (Quantifoil Au R1.2/1.3, 300 mesh), and flash frozen in liquid ethane using an FEI Vitrobot. The imaging conditions were kept identical as for the apo-state data. In brief, zero-energy loss images were recorded manually on an FEI Titan Krios microscope at 300 kV, using a slit width of 20 eV on a GIF-Quantum energy filter. A Gatan K2-Summit detector was used in super-resolution counting mode at a calibrated magnification of 35,714× (yielding a pixel size of 1.4 Å), and a dose rate of ~2.5 electrons/Å$^2$/s (~5 electrons/pixel/s). Exposures of 16 s were dose-fractionated into 20 movie frames. Defocus values in the DAPT-bound data set ranged from 0.6–2.8 μm.

### Image processing

Similar image processing procedures were employed for the apo-state and the DAPT-bound data sets. We used MOTIONCORR (*Li et al., 2013*) for whole-frame motion correction, CTFFIND4 (Rohou and Grigorieff) for estimation of the contrast transfer function parameters, and RELION-1.4 (*Scheres, 2012*) for all subsequent steps. References for template-based particle picking (*Scheres, 2015*) were obtained from 2D class averages that were calculated from a manually picked subset of the micrographs. A 20 Å low-pass filter was applied to these templates to limit model bias. All low-pass filters employed were cosine-shaped and fell to zero within 2 reciprocal pixels beyond the specified frequency. To discard false positives from the picking, we used initial runs of 2D and 3D classification to remove bad particles from the data. The selected particles were then submitted to 3D auto-refinement, particle-based motion correction and radiation-damage weighting (*Scheres, 2014*). The resulting 'polished particles' were used for masked classification with subtraction of the residual signal as described in the main text, and the original particle images from the resulting classes were submitted to a second round of 3D auto-refinement. All 3D classifications and 3D refinements were started from a 40 Å low-pass filtered version of the high-resolution consensus structure. Fourier Shell Coefficient (FSC) curves were corrected for the effects of a soft mask on the FSC curve using high-resolution noise substitution (*Chen et al., 2013*). Reported resolutions are based on gold-standard refinement procedures and the corresponding FSC=0.143 criterion

(*Scheres and Chen, 2012*). Prior to visualization, all density maps were corrected for the modulation transfer function (MTF) of the detector, and then sharpened by applying a negative B-factor that was estimated using automated procedures (*Rosenthal and Henderson, 2003*).

For the apo-state data set, the template-based algorithm picked 1.8 million particles from 2,925 micrographs, and 412,272 particles were selected after initial 2D and 3D classification. Subsequent 3D auto-refinement and particle polishing yielded a 3.5 Å map with fuzzy densities in the transmembrane region. Masked classification into eight classes with subtraction of the residual signal yielded three classes with good density as described in the main text. Poor reconstructed density was observed in the other five classes. Separate 3D auto-refinements of the corresponding particles in the original data set for the three best classes gave rise to reconstructions to 4.0– 4.3 Å resolution (also see *Figures 2–3*, *Table 1*).

For the DAPT-bound state, 1.4 million particles were picked from 2,206 micrographs, and initial classification selected 271,361 particles. After particle polishing, this subset gave rise to a 4.3 Å resolution map with relatively poor density in the transmembrane domain. Application of the masked classification procedure with residual signal subtraction into eight classes identified a single class with good density. After 3D auto-refinement, the corresponding 51,366 particles gave a map with a resolution of 4.2 Å, which showed improved density in the transmembrane domain.

## Instructions for masked classification with subtraction of residual signal

To expedite application of the modified classification procedures proposed in this paper by others, we describe these steps in more detail. The mask for masked classification on the PS1 subunit (the red part of the signal in *Figure 1*) was generated by converting the atomic model of the PS1 subunit from the high-resolution consensus structure (including a poly-alanine model for TM2) into a density map using the program `e2pdb2mrc.py` from EMAN2 (*Tang et al., 2007*). This map was then converted into a soft-edged mask using `relion_mask_create`. A mask around the entire γ-secretase complex, including the belt of fuzzy density from the amphipols, was generated using standard auto-masking from the RELION post-processing procedure. Subtraction of the PS1 mask from the mask of the entire complex using `relion_image_handler` yielded a mask containing only nicastrin, Aph-1, Pen-2 and the amphipol belt (the yellow part of the signal in *Figure 1*). This mask was applied to the 3.5 Å map that was calculated from a consensus refinement using all 400 thousand selected apo-state particles. The resulting masked map (`yellow.mrc`) was used for subtraction of the signal from the experimental particles as outlined in *Figure 1*. The corresponding program has been available in RELION from release 1.3 onwards and is used as follows:

```
relion_project --i yellow.mrc --subtract_exp --angpix 1.4
--ctf --ang Refine3D/run1_data.star --o newparticles
```

The `Refine3D/run1_data.star` file was produced by the consensus refinement and contains the orientation and CTF parameters of all 400 thousand particles. The command generates a new particle image stack called `newparticles.mrcs` and a new STAR-file with all relevant metadata called `newparticles.star`. The latter is used directly as input in the masked classification run, which may be launched from the RELION GUI using standard inputs, and providing the mask around the PS1 subunit as 'Reference mask' on the Optimisation tab. Because of the small size of the PS1 subunit, we chose to set the 'Perform image alignment' option on the Sampling tab to 'No'.

## Simulation of particle images

To generate the simulated particle images described in *Figure 2—figure supplement 4*, we also used the `relion_project` program:

```
relion_project --i Refine3D/run1_class001.mrc --o
simulated_run1 --ctf --angpix 1.4 --ang
Refine3D/run1_data.star --add_noise --model_noise
Refine3D/run1_model.star
```

By providing the data.star file (option `--ang`) and final map (option `--i`) from the refinements of the classes described in *Figure 3*, we generated simulated particles with similar orientational distributions and CTF parameters as those observed for the experimental data. By using the estimated power spectra of the noise for the experimental images (as provided through the `--model_noise` option), also the simulated spatial frequency-dependent signal-to-noise ratios are similar to those in the experimental data.

## Atomic modeling

Model building for the three apo-state classes and the DAPT-bound structure was started from the coordinates that were built in our 3.4 Å apo-state consensus structure (PDB ID: 5A63). Nearly all of the residues from Aph-1 and nicastrin fitted well into the four maps, but parts of PS1 and Pen-2 had to be manually adjusted in COOT (*Emsley et al., 2010*). Building of TM2 and the lower part of TM6 in PS1 was started from idealized α-helices, and sequence assignment of TM2 was guided by comparison with the structure of PSH (PDB ID: 4HYC) and by recognizable side chain features for Phe and Tyr residues. All models were refined in REFMAC (*Murshudov et al., 1997*), using modified procedures for cryo-EM maps (*Brown et al., 2015*) and secondary structure restraints generated by ProSMART (*Nicholls et al., 2014*). Overfitting of the atomic model for the DAPT-bound structure was monitored by refining the model in one of the half-maps from the gold-standard refinement approach, and testing the resulting model against the other half-map (*Amunts et al., 2014*). The same relative weight between the EM-density and geometric terms that resulted in good fits to the density without overfitting for the DAPT-bound structure was used for the final refinement of all four structures.

## Mass spectrometry

Protein samples (5 μg purified γ-secretase) were resolved by sodium dodecyl sulphate polyacrylamide gel electrophoresis (SDS-PAGE) on 4–12% Bis-Tris gels (Life Technologies, Carlsbad, CA) using MES running buffer (Formedium, UK) for 32 min at 200V. The gel was stained with InstantBlue protein stain (Expedeon, San Diego, CA) for direct protein visualization. Gel slices were prepared for mass spectrometric analysis using the Janus liquid handling system (PerkinElmer, UK). Briefly, the excised protein gel pieces were placed in a well of a 96-well microtitre plate and destained with 50% v/v acetonitrile and 50 mM ammonium bicarbonate, reduced with 10 mM DTT, and alkylated with 55 mM iodoacetamide. After alkylation, proteins were digested with 6 ng/μl trypsin (Promega, UK) overnight at 37°C. The resulting peptides were extracted in 2% v/v formic acid, 2% v/v acetonitrile. The digest was analysed by nano-scale capillary LC-MS/MS using an Ultimate U3000 HPLC (ThermoScientific Dionex, San Jose, USA) to deliver a flow of approximately 300 nl/min. A C18 Acclaim PepMap100 5 μm, 100 μm x 20 mm nanoViper (ThermoScientific Dionex, San Jose, USA), trapped the peptides prior to separation on a C18 Reprosil-pur 3 μm, 75 μm x 105 mm PicoCHIP (New Objectives, MA, USA). Peptides were eluted with a gradient of acetonitrile. The analytical column outlet was directly interfaced with a hybrid linear quadrupole fourier transform mass spectrometer (LTQ Orbitrap XL, ThermoScientific, San Jose, USA). Data-dependent analysis was carried out, using a resolution of 30,000 for the full MS spectrum, followed by five MS/MS spectra in the linear ion trap. LC-MS/MS data were then searched against a protein database (UniProt KB) using the Mascot search engine programme (Matrix Science, UK) (*Perkins et al., 1999*). Database search parameters were set with a precursor tolerance of 10 ppm and a fragment ion mass tolerance of 0.8 Da. One missed enzyme cleavage was allowed and variable modifications for oxidized methionine, carbamidomethyl cysteine, pyroglutamic acid, phosphorylated serine, threonine and tyrosine were included. MS/MS data were validated using the Scaffold programme (Proteome Software Inc., USA) (*Keller et al., 2002*). All data were additionally interrogated manually.

## Western blotting

Protein samples (1.6 μg purified γ-secretase) were resolved by SDS-PAGE on 4–12% Bis-Tris gels (Life Technologies) using MES running buffer (Formedium) for 32 min at 200V. The gel was transferred to Immobilon-P 0.45 μm PVDF membrane (Millipore, Germany) in 1X transfer buffer (Life Technologies) supplemented with 20% methanol for 1 hr at 65V. The membrane was subsequently blocked for 1 hr at room temperature in 5% bovine serum albumin/Tris-buffered saline and Tween-

20 (TBST) prior to incubation with indicated primary antibodies (anti-TMP21, ab133771, 1:500, Abcam, UK; anti-VAMP-8, ab76021, 1:5000 Abcam, UK; anti-Miner1, 13318-AP, 1:500, Proteintech, UK) overnight at 4°C. The membrane was washed with TBST and incubated with horseradish peroxidase-linked goat anti-rabbit IgG (NA934VS, GE Healthcare) for 1 hr at room temperature. The membrane was washed extensively with TBST and target proteins detected on FUJI Medical X-Ray Super RX film (100 NIF 18 x 24, Fujifilm, UK) using Amersham ECL Western Blotting Detection Reagent (GE Healthcare, UK).

## Acknowledgements

We thank Peilong Lu and Dan Ma for contributions to γ-secretase purification; Shaoxia Chen and Christos Savva for support with electron microscopy; Jake Grimmett and Toby Darling for support with high-performance computing; and Sarah Maslen and Mark Skehel for support with mass spectrometry. This work was supported by funds from the Ministry of Science and Technology (2014ZX09507003006 to Y.S.), the National Natural Science Foundation of China (31130002 and 31321062 to Y.S.), a European Union Marie Curie Fellowship (to X.-C.B.), and the UK Medical Research Council (MC_UP_A025_1013, to S.H.W.S.).

## Additional information

### Competing interests

SHWS: Reviewing editor, *eLife* The other authors declares that no competing interests exist.

### Funding

| Funder | Grant reference number | Author |
|---|---|---|
| Medical Research Council | MC_UP_A025_1013 | Sjors HW Scheres |
| Ministry of Science and Technology of the People's Republic of China | 2014ZX09507003006 | Yigong Shi |
| National Natural Science Foundation of China | 31130002 and 31321062 | Yigong Shi |
| European Commission | Marie Curie Fellowship | Xiao-chen Bai |

The funders had no role in study design, data collection and interpretation, or the decision to submit the work for publication.

### Author contributions

X-CB, Collected and processed cryo-EM data and performed atomic model refinement, Acquisition of data, Analysis and interpretation of data, Drafting or revising the article; ER, Performed Western blot analysis, Acquisition of data, Drafting or revising the article; GY, Purified γ-secretase, Contributed unpublished essential data or reagents; YS, Initiated the γ-secretase structure determination project, Drafting or revising the article; SHWS, Developed the new classification approach, supervised the work presented, Conception and design, Analysis and interpretation of data, Drafting or revising the article

### Author ORCIDs

Sjors HW Scheres, http://orcid.org/0000-0002-0462-6540

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
