## [Decision Letter]

Thank you for submitting your work entitled "Sampling the conformational space of the catalytic subunit of human γ-secretase" for consideration by *eLife*. Your article has been reviewed by two peer reviewers, and the evaluation has been overseen by a Reviewing Editor and John Kuriyan as the Senior Editor.

The reviewers have discussed the reviews with one another and the Reviewing editor has drafted this decision to help you prepare a revised submission.

Summary:

This is a well-written manuscript that adds a further useful step in the process of single-particle reconstruction, especially where there is significant conformation and/or compositional heterogeneity. The central idea is to mask the background in a way that also takes advantage of the knowledge of the orientation of each particle in the projection image, leading to a more accurate way to carry out localized or "focused" refinement. Nevertheless, both peer reviewers query the novelty of the approach and have a number of other concerns, some of them substantial, which would need to be addressed in a revised manuscript.

Essential revisions:

1) The central idea of the manuscript has been around for a while. It is surprising that the authors do not cite papers which have taken a similar approach to subtraction of residual signal in earlier work, especially studies of viruses by cryo-EM. See for example:

"Structural analysis of viral nucleocapsids by subtraction of partial projections", Ying Zhang, Victor A. Kostyuchenko, and Michael G. Rossmann

http://www.ncbi.nlm.nih.gov/pmc/articles/PMC1876683/

and:

"Bacteriophage phi29 scaffolding protein gp7 before and after prohead assembly", Marc C Morais et al.

http://www.nature.com/nsmb/journal/v10/n7/full/nsb939.html

Most likely Ludtke et al. have also described something similar many years ago in the context of studying GroEL by EMAN.

2) The authors refer to their methodology as "classification", while what they clearly mean is "clustering". This makes Figure 1 confusing, as the scheme is most likely a part of the iterative refinement process. The term classification is generally applied to algorithms for the assignment of items to given templates when the templates themselves and their numbers are known. What the authors do is clearly clustering, which normally describes a procedure which explores data structure, to detect natural groupings without a priori knowledge of group templates. This confusion in the EM field is not new, but instead of continuing the tradition, the authors might consider breaking with it or at least to specify clearly what they mean.

3) The method described in this manuscript appears to be based on a fundamental misunderstanding of the original design of the "focused classification", as originally developed by Penczek, Frank and Spahn, JSB 2006 (see Figure 2 and associated text in this paper). Moreover, Bai et al. claim that Penczek and Frank overlooked a basic inconsistency of the procedure, a notion that is challenged by one reviewer.

4) The method is not tested with simulated data. Instead of using only one experimental data set, it would be better to establish the general principles by using data sets derived from a known structure.

5) Unless the authors employ a non-linear projection algorithm, four operations in the upper left corner of Figure 1 commute. In other words, it makes no difference whether projections of two structures are subtracted from each other or structures themselves are subtracted and the difference projected. Possibly the impression that the figure implies otherwise stems from an imprecise description of the design.

6) Another main concern relates to the lack of biochemical purity of the γ-secretase sample. The authors describe an extra helix in 2 of the "apo" classes which they attribute to a co-purified protein, and they were even able to biochemically identify 4 extra proteins that were present in the sample. This raises the question whether the other state they see also has an extra protein bound to the γ-secretase complex and how much these contaminating proteins are influencing the "apo" γ-secretase structure they report both in this manuscript and in their earlier Nature paper. Similarly, because the density for the ligand is so poor in the 4.2Å map, it seems possible that the density they attribute to the inhibitor is actually a piece of one of the extra proteins in their sample. The reviewers wonder if the authors tried the same focused classification procedure on the liganded state that they used on the "apo" state, and whether that would give any insight into whether the "ligand" density they see is actually from the inhibitor or from one of the extra proteins.

---

## [Author Response]

Essential revisions:

*1) The central idea of the manuscript has been around for a while. It is surprising that the authors do not cite papers which have taken a similar approach to subtraction of residual signal in earlier work, especially studies of viruses by cryo-EM. See for example:*

"Structural analysis of viral nucleocapsids by subtraction of partial projections", Ying Zhang, Victor A. Kostyuchenko, and Michael G. Rossmann

http://www.ncbi.nlm.nih.gov/pmc/articles/PMC1876683/

and: "Bacteriophage phi29 scaffolding protein gp7 before and after prohead assembly", Marc C Morais et al.

*http://www.nature.com/nsmb/journal/v10/n7/full/nsb939.html Most likely Ludtke et al. have also described something similar many years ago in the context of studying GroEL by EMAN.*

We apologize for the lack of a better historical perspective in our original manuscript. To give credit to the authors mentioned by the reviewers (and two other recent papers), we have added the following text to the Discussion: "The idea to subtract part of the signal from experimental cryo-EM images is not new. […] Its successful application to γ-secretase demonstrates its potential for complexes that are considered to be relatively small for cryo-EM structure determination, and shows that it allows separation of protein structures that differ only in the orientation and position of a few α-helices".

In addition, we have removed all claims of novelty from the manuscript.

*2) The authors refer to their methodology as "classification", while what they clearly mean is "clustering". This makes Figure 1 confusing, as the scheme is most likely a part of the iterative refinement process. The term classification is generally applied to algorithms for the assignment of items to given templates when the templates themselves and their numbers are known. What the authors do is clearly clustering, which normally describes a procedure which explores data structure, to detect natural groupings without a priori knowledge of group templates. This confusion in the EM field is not new, but instead of continuing the tradition, the authors might consider breaking with it or at least to specify clearly what they mean.*

The reviewers are correct that the term 'clustering' is generally used for unsupervised approaches in pattern recognition, whereas the term classification is used for supervised approaches. However, as also pointed out by the reviewers, in cryo-EM one often speaks of unsupervised classification, when actually referring to a clustering approach. Also in the RELION interface, the method is referred to as '3D classification'. Therefore, to avoid confusion of the most likely users of the method, we chose to still refer to our method as masked 'classification' with signal subtraction. However, to avoid confusion for those readers from the wider fields of pattern recognition, we included the following explicit definition of our terminology in the revised manuscript: "Similarly, masked multi-reference refinement may be used as a clustering tool, i.e. to separate experimental particle images based on differences in a defined region of interest. We refer to this approach as masked 3D classification."

*3) The method described in this manuscript appears to be based on a fundamental misunderstanding of the original design of the "focused classification", as originally developed by Penczek, Frank and Spahn, JSB 2006 (see Figure 2 and associated text in this paper). Moreover, Bai et al. claim that Penczek and Frank overlooked a basic inconsistency of the procedure, a notion that is challenged by one reviewer.*

Because the reviewers do not see clear parallels with the focused classification method described by Penczek, Frank and Spahn, we have removed this comparison from the manuscript. Instead, we now compare the approach with our previous approach of (multi-reference) masked refinements without signal subtraction.

*4) The method is not tested with simulated data. Instead of using only one experimental data set, it would be better to establish the general principles by using data sets derived from a known structure.*

We have now included results on a simulated data set as a new supplement to Figure 2. In the main text, we mention "In addition, we tested our method on a simulated set of images containing a mixture of projections from the maps of classes 1 and 2 in Figure 3. For these simulations we used similar signal-to-noise ratios, CTF parameters and orientational distributions as observed in our experimental data set (also see Materials and methods). Masked classification with signal subtraction on the PS1 subunit correctly identified 93% of the simulated particles (Figure 2—figure supplement 4)".

In addition, we describe our method of simulating data with realistic signal-to-noise ratios in more detail in the Materials and methods section, as we think this may also be a useful tool for others.

5) Unless the authors employ a non-linear projection algorithm, four operations in the upper left corner of Figure 1 commute. In other words, it makes no difference whether projections of two structures are subtracted from each other or structures themselves are subtracted and the difference projected. Possibly the impression that the figure implies otherwise stems from an imprecise description of the design.

The reviewers are correct in observing that the four left panels in Figure 1 commute. However, the lower-left panel is a 2D experimental image, which will need to be compared with a 2D projection of a 3D reference structure in order to assign its relative orientation and/or class. The upper-left panel is an estimate of the signal in the experimental image. Therefore, although one could indeed subtract 3D maps from each other, one needs a projection step to go to the 2D space of the experimental data. To make it clearer that the lower-left panel is experimental data (and that one thus has no direct access to the underlying 3D structure that it represents), we have removed the vertical arrow on the left side of the original Figure, and increased the size of the label of the experimental image.

*6) Another main concern relates to the lack of biochemical purity of the γ-secretase sample. The authors describe an extra helix in 2 of the "apo" classes which they attribute to a co-purified protein, and they were even able to biochemically identify 4 extra proteins that were present in the sample. This raises the question whether the other state they see also has an extra protein bound to the γ-secretase complex and how much these contaminating proteins are influencing the "apo" γ-secretase structure they report both in this manuscript and in their earlier Nature paper. Similarly, because the density for the ligand is so poor in the 4.2Å map, it seems possible that the density they attribute to the inhibitor is actually a piece of one of the extra proteins in their sample. The reviewers wonder if the authors tried the same focused classification procedure on the liganded state that they used on the "apo" state, and whether that would give any insight into whether the "ligand" density they see is actually from the inhibitor or from one of the extra proteins.*

We have included the results of the masked classification approach on the liganded state data set as Figure 5—figure supplement 2 in the revised manuscript, and added the following statement to the main text: "For these data, masked classifications with signal subtraction revealed only a single class with good density for the transmembrane helices, and this class did not show any helical-like density in the cavity between TM2, TM3 and TM5 of PS1 (Figure 5—figure supplement 2)".